# Assessment of Global Longitudinal and Circumferential Strain Using Computed Tomography Feature Tracking: Intra-Individual Comparison with CMR Feature Tracking and Myocardial Tagging in Patients with Severe Aortic Stenosis

**DOI:** 10.3390/jcm8091423

**Published:** 2019-09-10

**Authors:** Emilija Miskinyte, Paulius Bucius, Jennifer Erley, Seyedeh Mahsa Zamani, Radu Tanacli, Christian Stehning, Christopher Schneeweis, Tomas Lapinskas, Burkert Pieske, Volkmar Falk, Rolf Gebker, Gianni Pedrizzetti, Natalia Solowjowa, Sebastian Kelle

**Affiliations:** 1Department of Internal Medicine/Cardiology, German Heart Center Berlin, 13353 Berlin, Germany; 2Department of Cardiology, Medical Academy, Lithuanian University of Health Sciences, 50161 Kaunas, Lithuania; 3Philips Healthcare, 22335 Hamburg, Germany; 4Klinik für Kardiologie und Internistische Intesivmedizin, Krankenhaus der Augustinerinnen, 50678 Köln, Germany; 5DZHK (German Centre for Cardiovascular Research), Partner Site Berlin, 10785 Berlin, Germany; 6Department of Internal Medicine/Cardiology, Charité Campus Virchow Clinic, 13353 Berlin, Germany; 7Department of Cardiothoracic Surgery, German Heart Center Berlin, 13353 Berlin, Germany; 8Department of Engineering and Architecture, University of Trieste, 34127 Trieste, Italy

**Keywords:** systemic disease, cardiac computed tomography, cardiac magnetic resonance, feature tracking, tagging, myocardial deformation, strain

## Abstract

In this study, we used a single commercially available software solution to assess global longitudinal (GLS) and global circumferential strain (GCS) using cardiac computed tomography (CT) and cardiac magnetic resonance (CMR) feature tracking (FT). We compared agreement and reproducibility between these two methods and the reference standard, CMR tagging (TAG). Twenty-seven patients with severe aortic stenosis underwent CMR and cardiac CT examinations. FT analysis was performed using Medis suite version 3.0 (Leiden, The Netherlands) software. Segment (Medviso) software was used for GCS assessment from tagged images. There was a trend towards the underestimation of GLS by CT-FT when compared to CMR-FT (19.4 ± 5.04 vs. 22.40 ± 5.69, respectively; *p* = 0.065). GCS values between TAG, CT-FT, and CMR-FT were similar (*p* = 0.233). CMR-FT and CT-FT correlated closely for GLS (*r* = 0.686, *p <* 0.001) and GCS (*r* = 0.707, *p <* 0.001), while both of these methods correlated moderately with TAG for GCS (*r* = 0.479, *p <* 0.001 for CMR-FT vs. TAG; *r* = 0.548 for CT-FT vs. TAG). Intraobserver and interobserver agreement was excellent in all techniques. Our findings show that, in elderly patients with severe aortic stenosis (AS), the FT algorithm performs equally well in CMR and cardiac CT datasets for the assessment of GLS and GCS, both in terms of reproducibility and agreement with the gold standard, TAG.

## 1. Introduction

Multiple systemic and neuromuscular diseases can affect the cardiovascular system at some point in their course. A wide variety of pathological processes fall under these definitions, some of which have pathognomonic cardiovascular manifestations [1]. However, non-specific manifestations, such as a subtle decline in regional or global myocardial function, are also common [2]. It can often go unnoticed until the ejection fraction (EF) starts to decline or clinical symptoms of heart failure begin to develop. Recently, myocardial strain has emerged as an imaging technique that adds information about myocardial function beyond the left ventricular ejection fraction (LVEF) [3]. Furthermore, recent studies have shown early reduction in myocardial strain in multiple systemic and neuromuscular disorders, such as amyloidosis [4], systemic sclerosis [5], rheumatoid arthritis [6], and Duchenne muscular dystrophy [7]. These data suggest that deformation imaging could become an important tool for the early identification of cardiac involvement in these patients. 

Due to its availability, speckle tracking echocardiography (STE) is the most widely used method for strain assessment. However, the accuracy and feasibility of STE is highly dependent on image quality [8], warranting the need for alternatives in certain patients. Cardiac magnetic resonance (CMR) not only allows for myocardial strain assessment, overcoming the shortcomings of echocardiography, but also offers tissue characterization ability that is second to none. Thus, it is an important tool in the diagnostic work-up of patients with systemic connective tissue disorders [9]. A recently developed cardiac magnetic resonance feature tracking (CMR-FT) technique has been validated against the gold standard myocardial tagging (TAG) and is now considered a preferred CMR tool for strain assessment [10]. The main advantage of CMR-FT is that it can be applied to steady-state free precession (SSFP) cine loops that are used in routine clinical practice, therefore not requiring additional image acquisition. Interestingly, although developed for CMR, the FT algorithm can also be applied to cardiac computed tomography (CT) datasets to assess myocardial strain [11,12]. Naturally, strain assessment from cardiac CT datasets has started to gain popularity. 

In this study, we used a single commercially available software solution to acquire global strain parameters via computed tomography feature tracking (CT-FT) and CMR-FT in a cohort of patients with severe aortic stenosis (AS). We compared agreement and reproducibility of global longitudinal (GLS) and global circumferential strain (GCS) between both these methods and the reference standard, TAG.

## 2. Experimental Section

### 2.1. Study Population

Twenty-six patients (14 females and 12 males, mean age 80.59 ± 5.87 years) with severe AS referred to our institution for transcathether aortic valve replacement (TAVR) were enrolled in this study. Further demographic and clinical data of the study population are listed in Table 1. AS was diagnosed and graded echocardiographically according to the latest European Society of Cardiology and European Association for Cardiothoraic Surgery guidelines [13]. All subjects underwent clinically indicated CMR and cardiac CT examinations. This study complies with the Declaration of Helsinki. Institutional Review Board approval was not necessary because it was a retrospective analysis of clinical data. According to local law, all individuals signed an informed consent form before entering the clinical CMR and cardiac CT. None of the observers could identify patient information when analyzing the data.

### 2.2. Cardiac Computed Tomography Acquisition

Contrast-enhanced, retrospectively electrocardiography (ECG)-gated cardiac scans were performed using a 2 × 128-slice multi-detector computed tomography scanner (Somatom Definition Flash, Siemens AG, Erlangen, Germany). The following study protocol was used: tube voltage 100, 120 kV, tube current 320 ref. mAs/rotation, rotation time 280 ms, slice collimation of 128 × 0.6 mm, with a temporal resolution of 75 ms, slice width of 0.75 mm, reconstruction increment of 0.4 mm, and reconstruction kernel B30f. Images were acquired in a cranio-caudal direction, from above the aortic sinuses to below the diaphragm. 

### 2.3. Cardiac Magnetic Resonance Acquisition

CMR acquisitions were made using a 1.5 Tesla magnetic resonance imaging (MRI) scanner (Achieva, Philips Healthcare, Best, The Netherlands). Signals were received using a five-element phased array cardiac coil. A four-lead vector ECG was used for R-wave triggering. A balanced steady-state free precession (bSSFP) sequence with breath hold was acquired in long-axis (LAX) two-, three-, and four-chamber views, as well as a short-axis (SAX) stack. This was used for volumetric and FT analysis. Acquisition parameters used were a repetition time (TR) of 3.3 ms, echo time (TE) of 1.6 ms, flip angle of 60°, acquisition voxel size of 1.8 × 1.7 × 8.0 mm^3^, and 30 phases per cardiac cycle. The complementary spatial modulation of magnetization (CSPAMM) technique was used to acquire tagging images in three short-axis planes (basal, medial, and apical) with a temporal resolution of 35 ms, spatial resolution of 1.4 × 1.4 mm, and a slice thickness of 8 mm. 

### 2.4. Cardiac CT Data Analysis

Original three-dimensional (3D) datasets were analyzed offline using the commercially available Medis Suite version 3.0 (Leiden, The Netherlands) software package to generate two-dimensional (2D) cine loops of three LAX slices (i.e., two-, three-, and four-chamber), three SAX slices (i.e., basal, mid, and apical), and a SAX stack with a slice thickness of 0.75 mm and a reconstruction increment of 0.4 mm. Images were generated with temporal resolution of 10 phases per cardiac cycle in 10% increments from early systole (0% cardiac cycle) to end-diastole (90% cardiac cycle). Care was taken to make sure that 2D cardiac CT reconstructions closely matched the anatomical locations of the images used for CMR analysis. End-systolic and end-diastolic cardiac phases were chosen visually. Endocardial and epicardial borders in the SAX stack were outlined manually to calculate the volumetric parameters, which were indexed to body surface area (BSA). Left ventricular mass index (LVMi), left ventricular end-diastolic volume index (LVEDVi), left ventricular end-systolic volume index (LVESVi), left ventricular stroke volume index (LVSVi), and left ventricular ejection fraction (LVEF) were calculated. Global longitudinal strain (GLS) was assessed by averaging the peak systolic strain values of 17 segments extracted from three LAX images, while global circumferential strain (GCS) was acquired from three SAX images using a 16-segment model.

### 2.5. CMR Data Analysis

bSSFP images were analyzed using Medis Suite version 3.0 (Leiden, The Netherlands) software in the same manner as cardiac CT images to determine LVMi, LVEDVi, LVESVi, LVSVi, LVEF, GLS, and GCS. Tagged images were analyzed using commercially available software Segment version 2.2 R6960. Endocardial and epicardial borders were manually outlined at an end-systolic timeframe in three short-axis slices (i.e., basal, mid, and apical). After applying an automatic propagation algorithm, quality of tracking was visually assessed, and changes were made as needed. GCS was derived using a 16-segment model by averaging the peak systolic values. TAG data of one of the subjects could not be analyzed due to breathing artefacts, therefore 26 patients were used for GCS comparisons. 

Due to the counter-intuitive increase of strain in more diseased subjects, we chose to report absolute values for easier interpretation. 

### 2.6. Statistics 

Data analysis was performed using commercially available software (GraphPad Prism 8, GraphPad Software, San Diego, CA, USA). The Shapiro–Wilk test was used to assess the normality of distribution of continuous variables. Unpaired Student’s *t*-test was used to compare differences between cardiac CT and MRI derived volumetric parameters and GLS. One-way ANOVA was used to compare differences in GCS between the three modalities. Pearson’s correlation coefficient and Bland–Altman analysis were used to assess inter-method agreement. Intra- and interobserver variability were assessed using two-way mixed intra-class correlation coefficient (ICC), Bland–Altman analysis, and coefficient of variance (CoV). This was defined as the standard deviation of the differences divided by the mean, in keeping with previous studies [14]. Agreement levels were defined according to previous studies [15] as follows: excellent if ICC > 0.74, good if ICC = 0.6‒0.74, fair if ICC = 0.4–0.59, poor if ICC < 0.4. *p*-values of <0.05 were considered statistically significant.

## 3. Results

### 3.1. Volumetric Assessment

Values of volumetric assessment are represented in Table 2. LVEDVi, LVESVi, LVSVi, LVEF, and LVMi values were similar between CMR and cardiac CT. There was excellent correlation between the two techniques in LVEDVi (*r* = 0.913, *p <* 0.001), LVESVi (*r* = 0.879, *p <* 0.001), LVEF (*r* = 0.791, *p <* 0.001), and LVMi (*r* = 0.971, *p <* 0.001), with good correlation for LVSVi (*r* = 0.619, *p <* 0.001). Results of the Bland–Altman analysis of volumetric measurements are shown in the figures (Figure 1a–d and Figure 2).

### 3.2. Strain Assessment

Figure 3 and Figure 4 show examples of GLS and GCS assessment in the same patient using different strain-assessment techniques. Strain values from each technique are represented in Table 3. GLS showed a trend towards being lower in CT-FT vs. CMR-FT (19.40 ± 5.04 vs. 22.40 ± 5.69, *p* = 0.065). GCS values were similar between all techniques (*p* = 0.233). There was good correlation between CMR-FT and CT-FT derived GLS (*r* = 0.686, *p <* 0.001) and GCS (*r* = 0.707, *p <* 0.001), while both of these methods had moderate correlation with TAG for GCS (*r* = 0.479, *p <* 0.001 for CMR-FT vs. TAG; *r* = 0.548 for CT-FT vs. TAG). Bland–Altman analysis revealed similarly wide limits of agreement (LOA) between all techniques in both GLS and GCS (Figure 5a–d and Table 4). 

### 3.3. Intraobserver and Interobserver Reproducibility 

The results of the reproducibility analyses are presented in Table 5. Intraobserver and interobserver agreement was excellent for all techniques. CMR-FT had worse intraobserver reproducibility for GLS (LOA ±2.4% vs. ±4.36%; CoV 6.8% vs. 10.1%) but performed better in interobserver comparison (LOA ±3.16% vs. ±5.5%; CoV 7.4% vs. 16.1%). TAG had superior reproducibility compared to the FT-based imaging technique for GCS, while FT-based techniques had similar results in interobserver and intraobserver comparisons. 

## 4. Discussion

### 4.1. Main Findings 

To our knowledge, this is the first study that compared CT-FT derived strain values to CMR-FT and TAG. The main findings of our study were as follows.
There was good correlation between CMR-FT and CT-FT for GLS and GCS assessment, while GCS derived from both CMR-FT and CT-FT had a moderate correlation with TAG;The intra- and interobserver reproducibility of CMR-FT and CT-FT were excellent;There were no significant differences between cardiac CT and CMR for the volumetric assessment of the LV.

In the past decade, multiple methods to assess myocardial strain parameters from cardiac CT datasets have emerged. Most use tissue tracking algorithms originally developed for CMR or echocardiography to track either endocardial or epicardial borders of the left ventricular (LV) in 2D cine loops generated from 3D cardiac CT datasets. As with CMR and echocardiography, these methods allow for the quantification of well-studied global strain parameters, GLS, GCS, and global radial strain (GRS). In 2010, Helle-Valle et al. used a multimodality tissue tracking algorithm, originally developed for analysis of echocardiographic images, to assess GRS in a cohort of ischemic heart disease patients (*n* = 20). They demonstrated that the results of this method had good correlation with GRS derived from TAG (*r* = 0.68) and that the method has the ability to discern scarred LV segments [16]. Buss et al. used a similar feature tracking algorithm in a cohort of congestive heart failure patients (*n* = 27) to obtain and compare global strain parameters from cardiac CT and transthoracic echocardiography datasets. They found close correlation for GRS (*r* = 0.97), GCS (*r* = 0.94), and GLS (*r* = 0.93) between these modalities [12]. In the largest study to date (*n* = 123), Fukui et al. compared FT-derived GLS in a cohort of severe AS patients and found moderate correlation between cardiac CT and transthoracic echocardiography (TTE) (*r* = 0.62) [11]. 

Another method, developed specifically for cardiac CT, allows for the quantification of a cardiac CT-specific 3D principal strain. First, images of neighboring phases are interpolated using a motion coherent algorithm to reduce noise and improve motion coherence. Interpolated images are then analyzed using image and model matching algorithms to create a 3D motion-vector matrix of the LV [17]. Voxels of interest can then be chosen within this matrix to derive either a regional or global principal strain. Unlike 2D strain parameters, 3D principal strain encompasses deformation in all directions. It, thus, incorporates longitudinal, circumferential, and radial components. It is expressed as a positive value [18]. In a recent study, Ammon et al. found a close correlation (*r* = −0.8) between 3D principal strain and STE-derived GLS in a cohort (*n* = 35) of severe AS patients [19]. 

As previously noted, in the present study, we used a commercially available feature tracking software to measure GLS and GCS from cardiac CT datasets and compared it to CMR-FT and TAG in a cohort of patients with severe AS. We found a strong correlation between CT-FT and CMR-FT, but CT-FT tended to underestimate both GLS (22.40 ± 5.69 vs. 19.4 ± 5.04, *p* = 0.065) and GCS (18.91 ± 5.97 vs. 18.13 ± 4.63, *p* = 0.233). Previous authors have noticed a similar underestimation when comparing cardiac CT-derived strain to STE [12,19]. This underestimation could be driven by low temporal resolution of cardiac CT-derived cine loops. As shown by Rösner et al., accuracy of STE-derived strain measurements is dependent on the temporal resolution of the recordings [20]. They found systematic underestimation of strain parameters at temporal resolutions of less than 30 frames/cardiac cycle. Indeed, our 2D cine cardiac CT reconstructions had a temporal resolution of 10 frames/cardiac cycle, while CMR cine loops were acquired at 30 frames/cardiac cycle. To our knowledge, the performance of feature tracking at lower temporal resolutions has not been investigated yet, thus further studies are needed. 

Interestingly, despite having a lower temporal resolution, when compared with TAG (the reference standard strain assessment technique), CT-FT and CMR-FT had similar correlation and LOA for GCS assessment. Additionally, reproducibility analysis revealed similar results for both FT-based techniques. If TAG data are taken as the ground truth, these findings suggest that FT algorithm performs equally well on both CMR and cardiac CT datasets. 

Additionally, due to its angle independency and high temporal and spatial resolutions, CMR is the gold standard technique for the functional assessment of the heart [21]. However, despite having the worse temporal resolution, cardiac CT has been shown to have a close correlation and good agreement with CMR for the assessment of LV volumes in multiple studies [22,23,24], indicating that these methods can be used interchangeably for volumetric assessment. Our results agree with these findings. 

### 4.2. Clinical Implications

There is an increasing number of patients who have implanted cardiac devices, and this decreases the feasibility of CMR due to potential artefacts or the inability to condition these devices. Our results imply that cardiac CT datasets are non-inferior to CMR datasets for the assessment of GLS and GCS using the FT algorithm. Furthermore, ours and multiple previous studies have shown potential interchangeability of volumetric measurements between cardiac CT and CMR. With further advancements in technology and an increase in temporal resolution, a cardiac CT might be used as a convenient follow-up tool to previous CMR assessments for both volumetric and strain measurements in patients with implanted cardiac devices. 

### 4.3. Limitations

Naturally, there are certain limitations in our study. Firstly, this was a small-scale single-center trial. Secondly, we only had TAG acquisitions for short-axis slices, therefore we could not compare FT-derived GLS to a reference standard imaging technique. However, previous studies suggest that CMR-FT has similar correlation and agreement to TAG for both GCS and GLS [25]. Thirdly, we did not have STE-derived strain parameters for a more comprehensive inter-modality comparison. Finally, given the retrospective nature of this trial and limited availability of high quality CT and CMR acquisitions that were made within a short timeframe in other populations, the trial was performed in a highly selected population of patients with severe AS. Thus, further studies are required to confirm these findings in more diverse cohorts. 

## 5. Conclusions

Our findings show that the FT algorithm performs equally well in both CMR and cardiac CT datasets for the assessment of GLS and GCS, both in terms of reproducibility and agreement with the gold standard, TAG. In the clinical routine, cardiac CT might be used as a convenient follow-up tool to previous CMR assessments for both volumetric and strain measurements in patients with implanted cardiac devices.

## Figures and Tables

**Figure 1 jcm-08-01423-f001:**
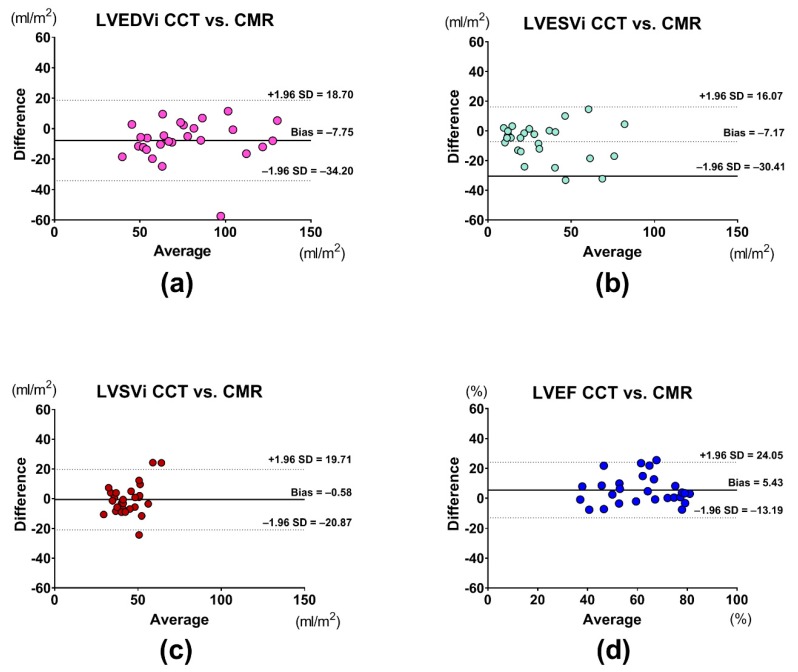
Bland–Altman analyses of (**a**) LVEDVi, (**b**) LVESVi, (**c**) LVSVi, and (**d**) LVEF assessment between CMR and CCT. Abbreviations: CCT: cardiac computed tomography; CMR: cardiac magnetic resonance; LVEF: left ventricular ejection fraction; LVEDVi: left ventricular end-diastolic volume index; LVESVi: left ventricular end-systolic volume index; LVSVi: left ventricular stroke volume index.

**Figure 2 jcm-08-01423-f002:**
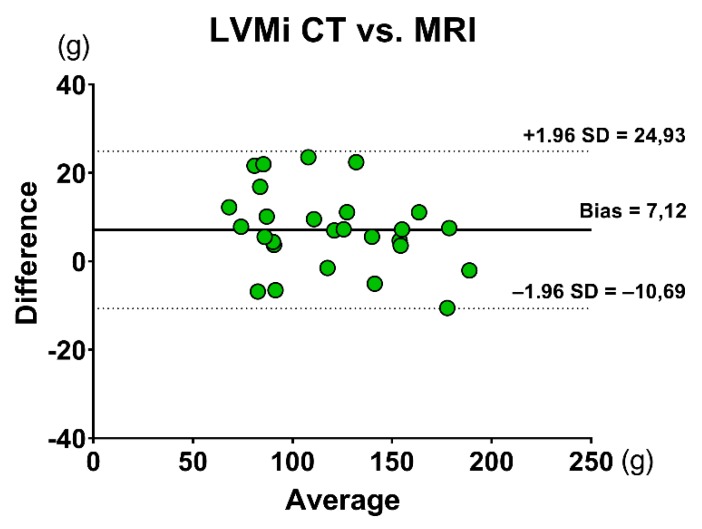
Bland–Altman analysis of LVMi assessment between CMR and CCT. Abbreviations: CCT: cardiac computed tomography; CMR: cardiac magnetic resonance; LVMi: left ventricular mass index.

**Figure 3 jcm-08-01423-f003:**
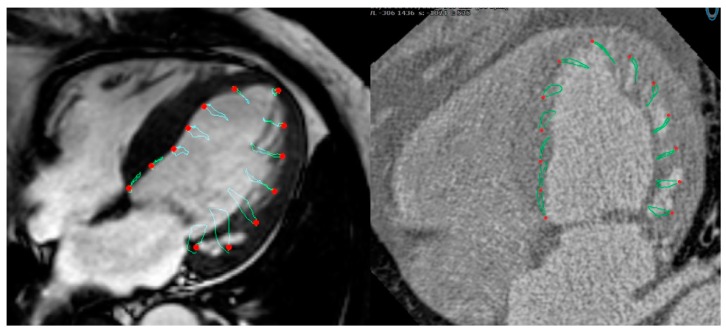
Assessment of GLS from the four-chamber long-axis (LAX) view in the same subject using CMR-FT (**left**) and CT-FT (**right**).

**Figure 4 jcm-08-01423-f004:**
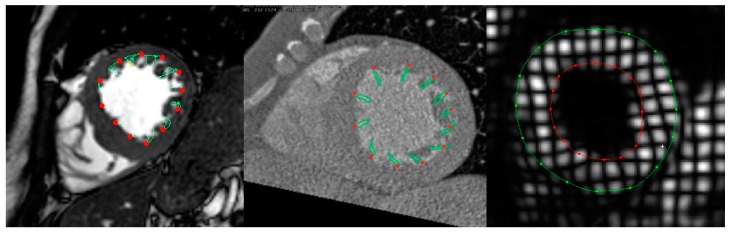
Assessment of GCS in the same subject from the mid-ventricular short-axis (SAX) view using CMR-FT (**left**), CT-FT (**middle**), and TAG (**right**).

**Figure 5 jcm-08-01423-f005:**
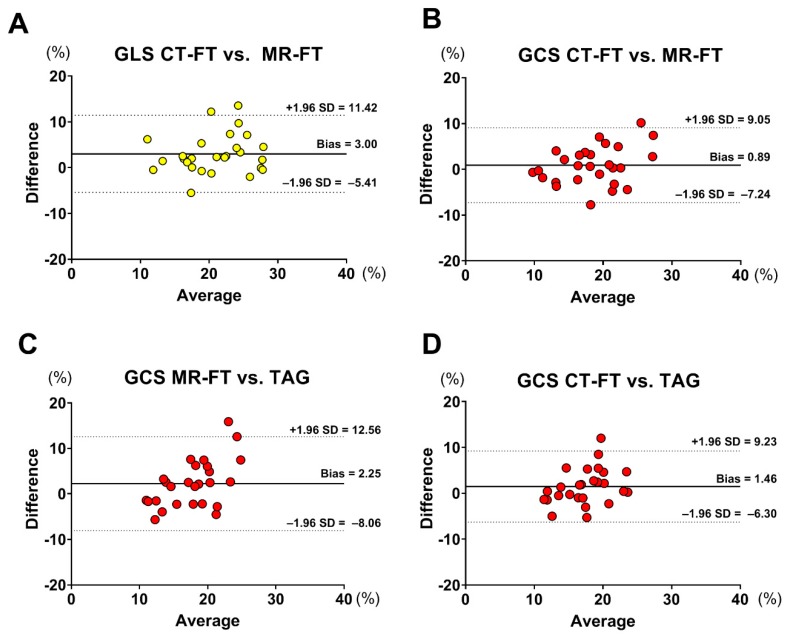
Bland–Altman analysis of the (**a**) GLS assessment between CT-FT and MR-FT; (**b**) GCS assessment between CT-FT and MR-FT; (**c**) GCS assessment between MR-FT and TAG; and (**d**) GCS assessment between CT-FT and TAG. Abbreviations: MR-FT: magnetic resonance feature tracking; CT-FT: computed tomography feature tracking; TAG: myocardial tagging; GLS: global longitudinal strain; GCS: global circumferential strain.

**Table 1 jcm-08-01423-t001:** Demographic and clinical data of the study population.

Variables.	*n* = 27 Mean ± SD or *n* (%)
Age	22.40 ± 5.69
Male	18.91 ± 5.97
Body mass index (kg/m^2^)	26.60 ± 3.60
Heart rate	67.59 ± 10.27
Clinical history	
Hypertension	25 (92.56%)
CAD	16 (59.25%)
Myocardial infarction	6 (22.22%)
History of CABG	5 (18.51%)
Stroke	4 (14.81%)
Diabetes mellitus type 2	6 (22.22%)
COPD	5 (18.51%)

Abbreviations: CAD: coronary artery disease; CABG: coronary artery bypass graft; COPD: chronic obstructive pulmonary disease.

**Table 2 jcm-08-01423-t002:** Values of volumetric assessment of the LV by CMR and CCT.

Measurement	CMR	CCT	*p*-Value
LVEF (%)	64.57 ± 14.55	59.15 ± 14.82	0.181
LVEDVi (mL/m^2^)	72.60 ± 27.22	80.35 ± 26.42	0.374
LVESVi (mL/m^2^)	28.62 ± 21.21	35.79 ± 23.39	0.293
LVSVi (mL/m^2^)	43.98 ± 11.65	44.56 ± 8.13	0.933
LVMi (g/m^2^)	62.13 ± 20.51	66.04 ± 19.42	0.471

Values are expressed as mean ± SD. Abbreviations: CCT: cardiac computed tomography; CMR: cardiac magnetic resonance; LVEF: left ventricular ejection fraction; LVEDVi: left ventricular end-diastolic volume index; LVESVi: left ventricular end-systolic volume index; LVMi: left ventricular mass index; LVSVi: left ventricular stroke volume index.

**Table 3 jcm-08-01423-t003:** Values of strain assessment of the LV by CMR and CCT.

Measurement	CMR-FT	CT-FT	TAG
GLS (%)	22.40 ± 5.69	19.4 ± 5.04	N/A
GCS (%)	18.91 ± 5.97	18.13 ± 4.63	16.66 ± 3.38

Values are expressed as mean ± SD. Abbreviations: LV: left ventricular; CCT: cardiac computed tomography; GLS: global longitudinal strain, GCS: global circumferential strain, CMR-FT: cardiac magnetic resonance feature tracking, CT-FT: computed tomography feature tracking; TAG: myocardial tagging.

**Table 4 jcm-08-01423-t004:** Tabular representation of Bland–Altman and Pearson’s correlation analyses for strain assessment.

Measurement	Comparison	Bias (%)	LOA (%)	Pearson’s R
GLS	CMR-FT vs. CT-FT	3.003	±8.415	0.6860
GCS GCS GCS	CMR-FT vs. CT-FT	0.888	±8.16	0.7067
	CMR-FT vs. TAG	2.250	±10.31	0.4799
	CT-FT vs. TAG	1.468	±7.77	0.5484

Abbreviations: GLS: global longitudinal strain; GCS: global circumferential strain; CMR-FT: cardiac magnetic resonance feature tracking; CT-FT: computed tomography feature tracking; LOA: limits of agreement; TAG: myocardial tagging.

**Table 5 jcm-08-01423-t005:** Reproducibility comparison of GLS and GCS between CMR-FT, CT-FT, and TAG.

	Bias (%)	Limits of Agreement (±)	CoV (%)	ICC (95% CI)
**Intraobserver reproducibility**				
**CMR-FT**				
GLS	0.09	4.36	10.1	0.960 (0.837–0.990)
GCS	−2.44	4.8	13.1	0.931 (0.439–0.985)
**CT-FT**				
GLS	−0.08	2.4	6.8	0.983 (0.932–0.996)
GCS	−0.05	5.0	14.4	0.949 (0.801–0.987)
**TAG**				
GCS	−0.08	1.26	3.9	0.992 (0.969–0.998)
**Interobserver reproducibility**				
**CMR-FT**				
GLS	0.03	3.16	7.4	0.982 (0.926–0.995)
GCS	−2.3	6.6	18.1	0.922 (0.629–0.981)
**CT-FT**				
GLS	1.25	5.5	16.1	0.866 (0.501–0.966)
GCS	0.39	5.1	14.6	0.940 (0.759–0.985)
**TAG**				
GCS	0.48	1.72	5.4	0.981 (0.918–0.995)

Abbreviations: GLS: global longitudinal strain; GCS: global circumferential strain; CMR-FT: cardiac magnetic resonance feature tracking; CT-FT: computed tomography feature tracking; TAG: myocardial tagging; CoV: coefficient of variance; ICC: intra-class correlation coefficient.

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
