# Peer review of "Assessment of Global Longitudinal and Circumferential Strain Using Computed Tomography Feature Tracking: Intra-Individual Comparison with CMR Feature Tracking and Myocardial Tagging in Patients with Severe Aortic Stenosis"

_jcm, 2019, doi:10.3390/jcm8091423_

Round 1

Reviewer 1 Report

Great research with aim to compare agreement and reproducibility of global longitudinal and circumferential strain both between CT and CMR, and reference standard TAG in 27 patients with severe aortic stenosis.

To improve the quality of article my offers are:

You should explain in the introduction or in the discussion why you choose patients with AS for the comparison the differences between CMR and CT methods to estimate GLS and GCS. What are the features in case of AS for GLS and GCS measurements comparing with healthy men? 

Author Response

Response to Reviewer 1 Comments

Point 1: You should explain in the introduction or in the discussion why you choose patients with AS for the comparison the differences between CMR and CT methods to estimate GLS and GCS.

Response 1: Given the fact that this was a retrospective trial, we chose patients with high quality CT and CMR acquisitions that were acquired within a short timeframe. During the time when these acquisitions were made, all of the patients who were referred for TAVI to our institution were examined by both CT and CMR as part of the standard protocol, thus this served as an ideal population for our trial. We have updated the manuscript accordingly and added the explanation in the limitations section. The changes can be found in pages 10-11, lines 274-277.

Point 2: What are the features in case of AS for GLS and GCS measurements comparing with healthy men?

Response 2: Previous studies that measured GLS and GCS in patients with AS and compared the results to healthy controls revealed that patients with AS tend to have reduced strain parameters1. However, our goal in this study was to compare the performance of strain imaging modalities to one another and we did not have a control group, thus we cannot comment on the specific differences in our study population.

  1. Al Musa T, Uddin A, Swoboda PP, et al. Myocardial strain and symptom severity in severe aortic stenosis: insights from cardiovascular magnetic resonance. Quant Imaging Med Surg. 2017;7(1):38-47. doi:10.21037/qims.2017.02.05

Reviewer 2 Report

Design: retrospective study in patients with aortic stenosis enrolled for TAVI.

The purpose of this paper is to assess global longitudinal (GLS) and circumferential strain (GCS) using cardiac computed tomography (CT) and cardiac magnetic resonance (CMR) feature tracking (FT), compared agreement and reproducibility both between these methods, and the reference standard CMR tagging (TAG).

Comments as follows:

General

I’m not sure that you can use as reference standard the TAG because TAG is a technique for evaluation of macroscopic kinetic alterations and not microscopic as strain.

Introduction

ok

Methods and Results

It not clear the clinical reason for CMR in patients enrolled for TAVI. Are you sure that this in a retrospective study?

Author Response

Response to Reviewer 2 Comments

Point 1: I’m not sure that you can use as reference standard the TAG because TAG is a technique for evaluation of macroscopic kinetic alterations and not microscopic as strain.

Response 1: Myocardial tagging was the first technique developed for myocardial strain assessment[1]. It has been improved upon multiple times since then and has remained the most validated strain imaging technique to date[2].  Given the fact that it is an acquisition-based strain imaging technique, it overcomes certain limitations of post-processing techniques, such as through-plane motion artifacts[1]. Furthermore, both speckle tracking echocardiography and feature tracking cardiac magnetic resonance were validated against tagging as the reference standard[3]. Additionally, recent publications that compare different CMR strain imaging modalities continue to use myocardial tagging as the reference[4]. For these reasons, we chose tagging as a reference modality to compare the performance of both CT and CMR feature tracking.

Point 2: It not clear the clinical reason for CMR in patients enrolled for TAVI. Are you sure that this in a retrospective study? 

 Response 2: Indeed, this was a retrospective study. During the time when these acquisitions were made all of the patients who were referred for TAVI to our institution would be examined by both CT and CMR as part of the standard protocol.

  1. Jiang K, Yu X. Quantification of regional myocardial wall motion by cardiovascular magnetic resonance. Quant Imaging Med Surg. 2014;4:345–57.
  2. Scatteia A, Baritussio A, Bucciarelli-Ducci C. Strain imaging using cardiac magnetic resonance. Heart Fail Rev. 2017;22:465–76. doi:10.1007/s10741-017-9621-8.
  3. Amzulescu MS, De Craene M, Langet H, Pasquet A, Vancraeynest D, Pouleur AC, et al. Myocardial strain imaging: review of general principles, validation, and sources of discrepancies. Eur Heart J Cardiovasc Imaging. 2019;20:605–19. doi:10.1093/ehjci/jez041.
  4. Cao JJ, Ngai N, Duncanson L, Cheng J, Gliganic K, Chen Q. A comparison of both DENSE and feature tracking techniques with tagging for the cardiovascular magnetic resonance assessment of myocardial strain. J Cardiovasc Magn Reson. 2018;20:26. doi:10.1186/s12968-018-0448-9.

Reviewer 3 Report

Dr Miskinyte and colleagues examined longitudinal and circumferential cardiac CT feature tracking vs those of CMR plus CMR tagging.

Comments:

The study population is that of severe aortic stenosis group awaiting TAVR, a highly selected population.  This should be made clear in the title and abstract. Given the selected population (severe AS with normal preserved LVEF), how would these results be applicable to other populations  There are no patient demographic details. There are minimal clinical details.

Author Response

Response to Reviewer 3 Comments

Point 1: The study population is that of severe aortic stenosis group awaiting TAVR, a highly selected population.  This should be made clear in the title and abstract.

Response 1: We do agree that results achieved in a highly selected cohort may not be representative of entire population, thus we have updated the title and the abstract accordingly.

Point 2: Given the selected population (severe AS with normal preserved LVEF), how would these results be applicable to other populations.

Response 2: Given the fact that this was a retrospective trial, this population was chosen due to availability of high quality CT and CMR data that was acquired within a short timeframe. We do acknowledge the fact that the results obtained in a highly selected population may not be representative of the entire population and further studies in both normal population and different pathological states are necessary to obtain the full picture. We have outlined this as a limitation in the updated manuscript, the changes can be found on page 10, lines 273-274.

Point 3: There are no patient demographic details.

Response 3: We have added the demographic details as in the updated manuscript under “study population” paragraph, page 2, lines 71-74 and as an additional table – Table 1.

Point 4: There are minimal clinical details.

Response 4: We have provided available clinical details of the cohort as an additional table – Table 1.

Round 2

Reviewer 2 Report

The revised manuscript reads better and the authors have addressed precisely all my comments.

Reviewer 3 Report

Most issues have been addressed.

In the abstract, suggest change "Our findings show that in patients with severe AoS...."  to "Our findings show that in elderly patients with severe AoS..."